# Evaluation of variability in performance and paw placement patterns by dogs completing the dog walk obstacle in an agility competition

**Juli K. DiMichele** [1], **Arielle Pechette Markley**[1], **Abigail Shoben**[2], **Nina R. Kieves** [1]*

**1** Department of Veterinary Clinical Science, College of Veterinary Medicine, The Ohio State University, Columbus, OH, United States of America, **2** Division of Biostatistics, College of Public Health, The Ohio State University, Columbus, OH, United States of America

* kieves.1@osu.edu

**Data Availability Statement:** All of the videos used in this study are publicly available at https://osf.io/24ex6/ (DOI: 10.17605/OSF.IO/24EX6).

## Abstract

The objective of this study was to describe paw placement patterns for canine athletes completing the dog walk obstacle during canine agility trials. It was hypothesized that dogs would demonstrate defined sets of paw placement patterns as they complete the dog walk obstacle and that those could be classified based on end contact behavior. Videos of 296 dogs attempting the dog walk obstacle at the 2021 UK Agility International (UKI) US Open were reviewed online. Data observed from video evaluation included front and rear limb paw placement across the dog walk and time to complete the obstacle. Results showed a high variability in obstacle performance. Mean time to complete the entire obstacle was 2.26 seconds (sd = 1.03). Mean and median completion times were qualitatively similar across all height classes. A slight majority of dogs hit the up ramp with their right foot first indicating running on their left lead (n = 185, 63%) with some variation observed between heights. Likewise, a slight majority (58%) of dogs hit the down ramp with their right front foot first (151/262). Given the high variation in completion times and paw placements, we could not identify clear patterns of dog walk performance. The large amount of variation observed with the dog walk obstacle suggests a need for future studies to employ alternative methods for objective gait analysis and to strategically select dogs to reflect the large variety in obstacle performance observed here.

## Introduction

Canine agility performance has exponentially grown in popularity over the last 10 years, and with it the need for objective information regarding performance kinematics as well as associated agility-related injuries. During canine agility trials, a handler directs a canine athlete through a set of obstacles while being timed. The goal of an agility trial is to perform all obstacles without making a mistake, or fault, in the least amount of time possible. Obstacles commonly observed in canine agility trials include the dog walk, seesaw, A-frames, tunnels, weave poles and bar jumps [1, 2]. Canine athletes performing agility are at risk for injuries due to the

**Funding:** The author(s) received no specific funding for this work.

**Competing interests:** The authors have declared that no competing interests exist.

chronic repetitive nature of agility training, as well as risk of acute injuries while performing at high speeds [1]. The most common injuries previously reported include the shoulder, iliopsoas muscle, neck and spine including the lumbosacral junction and sacroiliac joint [3–6]. The rate of injury reported in agility dogs is up to 42% [3–6], with 33% of dogs who undergo significant musculoskeletal injuries being unable to return to competition or returning to competition at lower heights [7].

Of the obstacles evaluated in canine agility trials, there has been minimal research regarding the dog walk obstacle, which is reportedly one of the more common obstacles associated with a higher risk of injury [4]. The dog walk obstacle requires the canine competitor to ascend one ramp, cross a flat center section and then descend another ramp to complete the obstacle. The top center section is flat and is consistently 48 inches above the ground. The three planks are 12 feet long and 12 inches wide and must be connected to minimize the risk of a dog's foot getting caught. The surfaces of the planks must be non-slip and slats are placed across the ramp sections to provide appropriate footing. Dogs must touch a contact zone with any part of one foot prior to exiting the obstacle otherwise they will receive a fault. The contact zone is confined to the lower portion of the down ramp and is distinguished from other parts of the ramp using a color specification, so it is easily identifiable by spectators and judges [2]. The exact length of the contact zone varies slightly between organizations (between 36 and 42 inches).

The requirement to touch the contact zone encourages handlers to train a specific end contact behavior. One common approach is a "2 on 2 off" contact behavior, where the dog is trained to fully stop at the end of the dog walk with their front paws on the ground and both rear feet in the contact zone on the down ramp. Another approach is to train a "running" contact behavior where the dog is trained to adjust their striding down across the dog walk to consistently have paws hit in the contact zone on the down ramp. Other approaches are reported less frequently [8], including a stop at the end of the down ramp with all 4 feet still on the obstacle and so-called "managed" contacts where the handler encourages the dog to slow down in order to have a paw touch the contact zone before continuing. While performing the obstacle, the dog is quickly alternating between uphill to downhill movements, which dictates them to change their pace and recruit different muscle groups in a matter of seconds. They must also quickly change from propulsive forces while ascending the up ramp to braking forces while descending on the down ramp.

Agility sports lack current data regarding definition of canine running styles, effective patterns for all obstacles and the long-term consequences for injury risk and the longevity of the canine athlete. A previous study [9] was able to clearly define paw placement pattern styles during completion of the weave pole obstacle, with five distinctive gait styles being identified.

To date, no evaluation of how dogs complete the dog walk obstacle has been performed. If distinctive running patterns are seen, as with the weave pole study, it could help plan for prospective analysis of dogs completing this obstacle and potentially determine risk factors for injury [9]. It would also help inform methodology of kinematic research studies regarding canine agility. Therefore, the objective of this study was to describe paw placement patterns for canine athletes completing the dog walk obstacle during canine agility trials. We hypothesized that dogs would demonstrate a number of identifiable paw placement patterns as they complete the dog walk obstacle.

## Materials and methods

All runs of the Last Chance Masters Agility course from the 2021 UK Agility International (UKI) U.S. Open were watched on a web video viewer (Youtube) in slow motion speed (0.25x). All videos were watched by a single primary reviewer (JKD). Reviewer consistency was

separately estimated by having two additional reviewers assess approximately 10% of runs. All data presented reflect the original observations of the primary reviewer to maximize the internal validity of the study. The videos of the competition were provided by 4LeggedFlix (https://4leggedflix.com/). The dog walk obstacle was the sixteenth obstacle to be performed and was completed after a curved tunnel obstacle. The dog would complete the curved tunnel and then head straight to the dog walk up ramp. The obstacle following the dog walk was a bar jump. The camera position was fixed on the right side of the dog and followed the dog for the length of the dog walk (Fig 1). The courses remained the same for each run and the camera angle was similar across all height categories; however, occasionally the human-operated camera would not move in time to observe the dog as it initially contacted the down ramp. As the video recordings reviewed were made by 4LeggedFlix for their coverage of the event, additional camera views were not available.

Specific data to be obtained regarding each agility run was pre-determined by the authors after an initial brief review of the video footage. Information obtained from the video about the time of the first contact with the dog walk, time to complete each section, and paw placement in each section was recorded using a Qualtrics survey. Times recorded were used by the timer provided by 4LeggedFlix in the YouTube video or with a handheld stopwatch (Apple, iPhone X). Completion time was defined by the time when the dog's first paw contacted the up ramp to the time when all four paws were no longer in contact with the obstacle. Time for the up ramp was recorded from time of initial contact to time of first contact with the middle ramp. Time for the middle ramp was recorded from time of first paw contact to time of first paw contact with the down ramp or time at which half the dog's body was over the down ramp. Time for the down ramp was from first paw contact (or half the dog's body over the down ramp) to the time when all four paws no longer in contact with the obstacle. For each ramp (up, middle, and down), the number of front paw hits (both front paws contacting the ramp) and rear paw hits was also recorded. Similarly, the first paw to contact each ramp was also recorded, as was the first paw to contact the ground after completing the obstacle. For the contact zone at the end of the down ramp, the number of paws that appeared to touch the contact zone was recorded. On the down ramp, the reviewer's judgement about if the dog appeared to slow down (or stop) was recorded, as was if the dog appeared to jump off the obstacle. If the dog appeared to stop, the location of the stop(s) were recorded, including if the dog fully stopped in the "2 on 2 off" position with its front feet on the ground and rear feet touching the obstacle. Additional observations recorded included which side of the ramp the handler was on during the obstacle completion, and if the dog completed the obstacle at the correct point of the course.

Breed and other signalment information was not available. UKI jump height category for all dogs was available from the online information for the event. Jump height categories are based on measured height at the withers, so dogs were divided into four height classes based on UKI jump height class gathered from the online data available. The four categories were <12.75", 12.75–17.5", >17.5–22", and >22".

Descriptive statistics (mean and median for time variables and percentages for categorical ones) by height category were used to assess overall performance and explore differences by height of the dog. Exploratory statistical tests for differences by height were performed to provide additional context for the descriptive statistics, with no adjustments made for multiple comparisons. These tests were done using a multivariate Wald test following linear regression with robust standard errors for mean completion time and using chi square tests for categorical paw strike variables.

Reviewer consistency was estimated as percent agreement and Krippendorf's alpha for categorical variables and the concordance correlation coefficient (CCC) for the continuous time

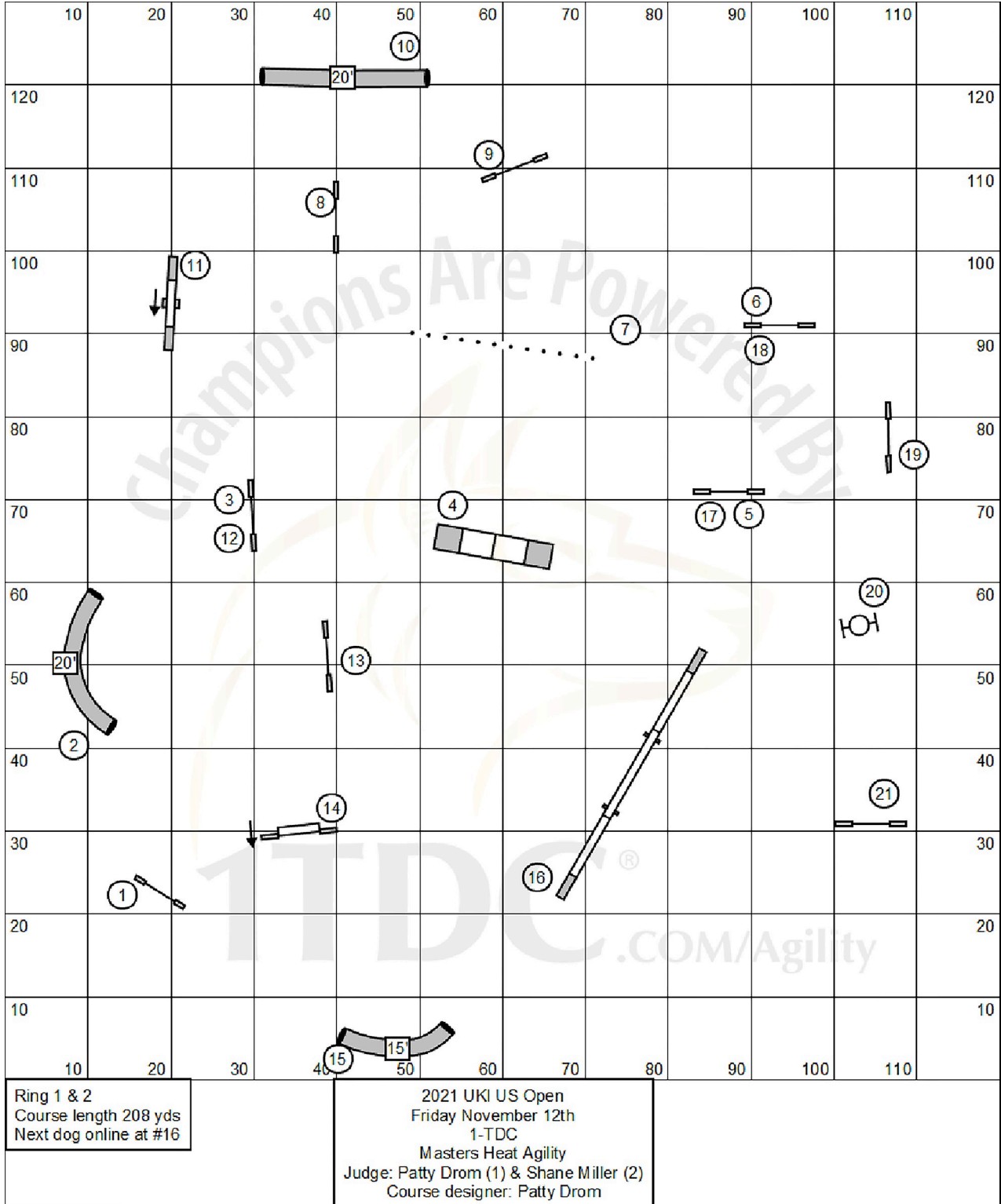

**Fig 1. Agility course layout.** Illustration of the agility course in the exact layout that was performed and observed by the current study. Image obtained directly from UKI International website: https://ukagilityinternational.com/wp-content/uploads/2021/11/Friday-print.pdf.

variables. All analyses were performed using Stata version 15.1 (StataCorp, College Station, TX).

## Results

A total of 296 dog walk attempts (at least one paw on the obstacle) were observed among all videoed runs of this course. Nearly all dogs (n = 283, 96%) took the obstacle in the flow of the course (after completing a u-shaped tunnel). Of the remaining 13 dogs, 11 attempted the dog walk after being resent to the u-shaped tunnel following a reset and 2 attempted the obstacle after a reset and without first completing the u-shaped tunnel. Similarly, nearly all dogs (n = 281, 95%) attempted the dog walk with the handler to the left side of the obstacle with the remaining dogs (n = 15) attempting the obstacle with the handler on the right side. Mean time to complete the entire obstacle was 2.26 seconds (sd = 1.03), with mean and median times similar across height classes (p = 0.70, Table 1). Height categories reflected the distribution of dogs competing at this event, with the most dogs in the >17.5–22" category (n = 129, 44%), followed by >22" (n = 81, 27%), 12.75–17.5" (n = 63, 21%), and <12.75" (n = 23, 8%).

A slight majority of dogs hit the up ramp with their right foot first indicating running on their left lead (n = 185, 63%) with taller dogs more likely to strike with their right foot first than shorter dogs (p = 0.001, Table 2). Likewise, a slight majority of dogs hit the down ramp with their right front foot first (n = 151; 58% among the 262 dogs observed). There was a difference by height of the dog (p = 0.002), with shorter dogs slightly more likely to hit with their left foot first. As expected, the total number of hits with the front and rear feet was lower for the taller dogs, particularly for the up and middle ramps. A majority of taller dogs completed the up and middle ramps with exactly two front and rear foot hits and a larger percentage of shorter dogs needed three (or more) hits to complete these two ramps (Table 2).

A variety of behaviors were observed around the contact zone on the down ramp. From video it appeared that 21% (n = 61) of dogs missed the contact zone completely with nearly all (93%, n = 57) of these dogs appearing to jump off the obstacle prior to the contact zone (1 dog fell and the other 3 were large dogs who appeared to run over the down ramp without touching the contact zone). Only 38 dogs (13%) fully stopped in the classic "2 on 2 off" position and this varied by height category, with the larger dogs more likely to stop in this position (Table 3, p = 0.066 for any difference in contact behavior by height). A true "running contact" type performance where the dog hit the contact and appeared to move through the contact zone at a consistent speed was observed for 77 dogs (26%), while the remaining 119 (40%) dogs touched the contact, but subsequently jumped off the obstacle. Most dogs appeared to run over the dog walk at a consistent speed (62%, n = 180, Table 4) and the remaining dogs either stopped

**Table 1. Dog walk performance times by height category.**

| | | | Height at the withers | | | | |
|---|---|---|---|---|---|---|---|
| | | All dogs (n = 296) | <12.75" (n = 23) | 12.75–17.5" (n = 63) | >17.5–22" (n = 129) | >22" (n = 81) | P for difference |
| Up Ramp | Mean (SD) | 0.63 (0.25) | 0.79 (0.31) | 0.67 (0.20) | 0.60 (0.28) | 0.59 (0.17) | 0.003 |
| | Median (25th, 75th) | 0.6 (0.5, 0.7) | 0.7 (0.6, 0.8) | 0.6 (0.5, 0.8) | 0.5 (0.5, 0.7) | 0.6 (0.5, 0.6) | |
| Middle Ramp | Mean (SD) | 0.76 (0.35) | 1.02 (0.52) | 0.82 (0.39) | 0.73 (0.31) | 0.68 (0.26) | 0.006 |
| | Median (25th, 75th) | 0.7 (0.6, 0.8) | 0.8 (0.7, 1.4) | 0.7 (0.6, 0.9) | 0.7 (0.6, 0.9) | 0.7 (0.5, 0.8) | |
| Down Ramp | Mean (SD) | 0.88 (0.72) | 0.71 (0.36) | 0.78 (0.73) | 0.92 (0.68) | 0.93 (0.83) | 0.098 |
| | Median (25th, 75th) | 0.6 (0.4, 1.2) | 0.6 (0.4, 1.0) | 0.5 (0.4, 0.9) | 0.7 (0.4, 1.3) | 0.5 (0.4, 1.2) | |
| Entire Obstacle | Mean (SD) | 2.26 (1.03) | 2.51 (1.12) | 2.27 (0.98) | 2.25 (1.01) | 2.21 (1.07) | 0.70 |
| | Median (25th, 75th) | 2.0 (1.5, 2.7) | 2.0 (1.7, 3.2) | 2.0 (1.6, 2.7) | 2.1 (1.5, 2.7) | 1.8 (1.5, 2.6) | |

**Table 2. Dog walk paw strikes by height category.**

| | All dogs (n = 296*) | <12.75 (n = 23) | 12.75–17.5 (n = 63) | >17.5–22 (n = 129) | >22 (n = 81) | P for difference |
|---|---|---|---|---|---|---|
| | | | **Height at the withers** | | | |
| | **All dogs (n = 296\*)** | **<12.75 (n = 23)** | **12.75–17.5 (n = 63)** | **>17.5–22 (n = 129)** | **>22 (n = 81)** | **P for difference** |
| First Paw to hit | | | | | | |
| Up ramp | | | | | | 0.001 |
| Right | 185 (63%) | 12 (52%) | 27 (44%) | 95 (74%) | 51 (63%) | |
| Left | 110 (37%) | 11 (48%) | 35 (56%) | 34 (26%) | 30 (37%) | |
| Not observed | 1 (0.3%) | 0 (0%) | 1 (2%) | 0 (0%) | 0 (0%) | |
| Down ramp | | | | | | 0.002 |
| Right | 151 (51%) | 8 (35%) | 21 (33%) | 81 (63%) | 41 (51%) | |
| Left | 111 (38%) | 9 (39%) | 33 (53%) | 37 (29%) | 32 (40%) | |
| Not observed | 33 (11%) | 6 (26%) | 9 (14%) | 10 (8%) | 8 (10%) | |
| Front foot hits (1 hit = both paws) | | | | | | |
| Up ramp | | | | | | <0.001 |
| <2 | 31 (10%) | 0 (0%) | 1 (2%) | 18 (14%) | 12 (15%) | |
| 2 | 194 (66%) | 5 (22%) | 40 (63%) | 89 (69%) | 60 (74%) | |
| 3 | 57 (19%) | 11 (48%) | 16 (25%) | 22 (17%) | 8 (10%) | |
| >3 | 14 (5%) | 7 (30%) | 6 (10%) | 0 (0%) | 1 (1%) | |
| Middle ramp | | | | | | <0.001 |
| <2 | 47 (16%) | 0 (0%) | 6 (10%) | 21 (16%) | 20 (25%) | |
| 2 | 128 (43%) | 10 (43%) | 27 (43%) | 58 (45%) | 33 (41%) | |
| 3 | 100 (34%) | 5 (22%) | 20 (32%) | 48 (37%) | 27 (33%) | |
| >3 | 21 (7%) | 8 (35%) | 10 (16%) | 2 (2%) | 1 (1%) | |
| Down ramp | | | | | | 0.001 |
| <2 | 67 (23%) | 0 (0%) | 9 (14%) | 35 (27%) | 23 (29%) | |
| 2 | 91 (31%) | 9 (39%) | 31 (49%) | 30 (23%) | 21 (26%) | |
| 3 | 58 (20%) | 8 (35%) | 13 (21%) | 22 (17%) | 15 (19%) | |
| >3 | 79 (27%) | 6 (26%) | 10 (16%) | 42 (33%) | 21 (26%) | |

*one dog fell off the dog walk prior to the down ramp so n = 295 for down ramp variables.

completely (18%, n = 52) or slowed down at least somewhat (20%, n = 59), with shorter dogs more likely to maintain a consistent speed (p = 0.010).

Performance on the down ramp varied more between dogs than did performance on the up and middle ramps. There was more variability in completion time (Table 1) and more variability in the number of paw hits (Table 2) even among dogs from the same height category on the down ramp. This variability was likely due to variability in contact behavior, with dogs who came to a complete "2 on 2 off" stop taking much longer on the down ramp than dogs with

**Table 3. Observed contact behavior (p for difference = 0.066).**

| | All dogs (n = 295) | <12.75 (n = 23) | 12.75–17.5 (n = 63) | >17.5–22 (n = 129) | >22 (n = 81) |
|---|---|---|---|---|---|
| | | | **Height at the withers** | | |
| | **All dogs (n = 295)** | **<12.75 (n = 23)** | **12.75–17.5 (n = 63)** | **>17.5–22 (n = 129)** | **>22 (n = 81)** |
| Missed contact | 61 (20.7%) | 3 (13.0%) | 20 (31.8%) | 22 (17.1%) | 16 (20.0%) |
| "2 on 2 off" stop | 38 (12.9%) | 0 (0.0%) | 4 (6.4%) | 19 (14.7%) | 15 (18.8%) |
| Running | 77 (26.1%) | 8 (34.8%) | 13 (20.6%) | 38 (29.5%) | 18 (22.5%) |
| Other* | 119 (40.3%) | 12 (52.2%) | 26 (41.3%) | 50 (38.8%) | 31 (38.8%) |

*Other behavior was touching the contact zone with at least one paw and subsequently jumping off the obstacle.

**Table 4. Observed deceleration on the down ramp (p for difference = 0.010).**

| | All dogs (n = 295) | Height at the withers | | | |
|---|---|---|---|---|---|
| | | <12.75 (n = 23) | 12.75–17.5 (n = 63) | >17.5–22 (n = 129) | >22 (n = 81) |
| Full stop | 52 (17.9%) | 0 (0.0%) | 7 (11.3%) | 28 (22.1%) | 17 (21.5%) |
| Slowing down | 59 (20.3%) | 3 (13.0%) | 8 (12.9%) | 31 (24.4%) | 17 (21.5%) |
| No change | 180 (61.9%) | 20 (87.0%) | 47 (75.8%) | 68 (53.5%) | 45 (57.0%) |

any other contact behavior (S1 Table). These dogs were also much more likely to have more than 3 front paw hits on the down ramp than dogs who performed another contact behavior (S2 Table). Dogs who performed the complete "2 on 2 off" behavior were also more likely to have more paw hits on the middle ramp than dogs who performed another contact behavior, although mean completion times were similar across all observed contact behaviors for the middle ramp.

Agreement between reviewers (across 41 unique dogs; not all dogs observed by all reviewers) was excellent for time variables (CCC>0.9 for all with mean differences <0.04 seconds). Agreement was also satisfactory or better (Krippendorf's alpha>0.7) for most categorical variables. There was somewhat less consistency among reviewers in determining if the dog jumped off the down ramp (Krippendorf's alpha = 0.5).

## Discussion

The objective of this study was to describe paw placement patterns for canine athletes completing the dog walk obstacle during canine agility trials. There was a large variation in observed dog walk obstacle performance, both in time to completion and in paw placement patterns. Our hypothesis was therefore rejected, as there was substantial variability in paw placement with no predominant patterns being observed in this population of agility athletes. Although no clear patterns were observed in the current study, the large amount of variation observed with the dog walk obstacle remains an important finding to inform further kinematic and kinetic research and correlate how the variability could relate to the development of injuries. The fact that we were unable to classify dogs based on dog walk performance contrasts with the prior Eicher et. al study that described five distinct paw placement styles in agility dogs performing the weave pole obstacle [9]. Differences in the ability to define paw placement styles in weave pole obstacles and not in the dog walk obstacle could be due to the different kinematics required by the dog to perform each obstacle. The weave pole agility obstacle requires the dog to move in both lateral and forward motions as they move between a series of poles. In contrast, the dog walk obstacle requires the dog to alternate between uphill to downhill movements. They must quickly change from propulsive forces while on the up ramp to braking forces while running or slowing down on the down ramp in order to place a foot within the contact zone.

The large variability in end of obstacle performance on the dog walk may also help explain why consistent patterns were not observed. While there are multiple training techniques for the weave obstacle, ultimately the completion of the obstacle is the same despite varying paw placement patterns. The dog walk obstacle has more variability in end performance behavior depending on whether the handler trains for a running, stopped, or other contact behavior. The results of the current study revealed that only 18% of dogs competing came to a full stop and that many dogs were jumping off the down ramp. This information suggests that a majority of dogs are not consistently performing true stopped contacts, nor are they performing true running contacts where the dog moves across the entire down ramp at a consistent speed. The

large number of dogs jumping off the down ramp was unexpected and suggests a need for future studies that include this behavior among the variety of approaches to the contact zone behavior.

There was a fair amount of variation in which paw hit the up ramp first, even though nearly all dogs would be expected to be on their left lead when approaching the dog walk, due to the curve of the tunnel and the presence of the majority of the handlers on the left side of the dog. This may suggest that some dogs are switching their lead leg as they approach the dog walk, in a way that they are preferentially hitting the dog walk with the same leg each time. This is consistent with a previous study by Appelgrein et. al., where it was demonstrated that most dogs entering the A-frame obstacle consistently landed with the same sided forelimb when contacting the A-frame over a series of nine trials [10]. If dogs are preferentially striking both the dog walk and the A-frame obstacle with the same forelimb, this finding may have implications for injury risk and evaluation based on knowledge of a specific dog's preferred leading limb.

In this study, there was a large degree of variation of paw placement observed both between and within height classes, particularly when performing the down ramp aspect of the dog walk. Many factors, including breed, conformation, speed while completing the obstacle, and training techniques may contribute to the high levels of variation seen in paw placement styles observed. In the study regarding paw placement in dogs performing weave pole obstacles, differences in the relative frequency of the various styles were observed between Border Collies and all other dog breeds even within the same height class [9]. These findings are important to note as differences in breed conformation and size may contribute to the variability observed in the present study, although it is unlikely to be the sole contributor. Additional objective gait analysis techniques are required to identify specific movement patterns during dog walk performance to determine the effect of differences between breeds, morphometric measurements, speed and contact behavior on movement patterns.

Previous studies have hypothesized that there may be an association between injury and performance of specific agility obstacles, naming the dog walk and A-frame obstacles as most often associated with injury based on handler-reported data [3–5]. The A-frame obstacle requires the dog to ascend and descend a ramp and, like the dog walk, there is a contact zone at the bottom of the down ramp that the dog must place a paw into to successfully complete the obstacle. Unlike the dog walk, the A-frame does not have a flat ramp in the middle of the two ramps and the angle of ascent and descent is steeper, though the obstacle is wider. Both obstacles require the dog to use propulsive forces up the ramp and braking forces down the ramp. Dogs enter both the A-frame and dog walk obstacle at an angle of incline that may act as a source of stress on the joints. A previous study evaluating kinematics of the carpus when entering the A-frame showed that hyperextension of the carpus occurs when dogs enter the a-frame [10], which may contribute to chronic repetitive injuries of the carpus. This study also evaluated whether carpal extension was decreased during A-frame performances when the A-frame was set at a lower incline, and revealed no differences in carpal range of motion [10] at various angles of incline. It is currently unknown how the degree of incline and decline of the dog walk affects kinematics.

An additional study investigating kinematics in working German shepherd dogs completing A-frame obstacles observed that most dogs jumped from the mid-point of the ramp to the ground [11], as these dogs are not trained to complete the obstacle in the same fashion as agility dogs and are not required to place a paw in the contact zone. Surprisingly, this observation was also common in the present study for dogs completing the dog walk, with a large percentage of dogs jumping from the down ramp of the obstacle rather than running off it. The act of jumping higher agility obstacles has been associated with more acute landing angles and an increase in peak vertical forces [11, 12] when landing, which may serve as another source of

stress on the joints leading to injuries while completing the a-frame and dog walk obstacles. Given the similarities observed between the two obstacles, further studies regarding paw placement styles of dogs completing the a-frame may be of interest to determine if there is a large amount of variability in paw placement occurring that could contribute to its association with injury and how it compares to the variability seen in dogs completing the dog walk obstacle.

Limitations of this study include the camera angle of the videos observed and the video quality used to observe the canine athletes. The single, non-fixed camera view followed the canine athlete as they crossed the dog walk obstacle and occasionally made it challenging for observation and interpretation of paw placements, most notably for the smaller height categories and for dogs who completed the obstacle very quickly. The quality of video on YouTube is another limitation when compared to other video programs that may have improved video quality and the ability to observe gait patterns frame by frame. The lack of consistent timing devices available in the Youtube videos occasionally required that the reviewer use a manual stopwatch to obtain completion time. This is a limitation because of the potential error inherently involved using a manual timer, however concordance of time measurements among reviewers, even when manually timed, was quite good. Additional limitations to the study included the lack of information available regarding the breed, sex, age and agility experience level of the dogs participating. Each dog was only observed performing the obstacle one time, leaving it unknown how consistent certain dogs are in their paw placement patterns from dog walk to dog walk. The event observed was a national event, which does not account for dogs performing at lower levels of agility and presents a selection bias for dogs performing at higher levels of agility. Because of the higher stakes involved in performing in a national event, handlers may intentionally attempt to increase the speed of their dog, thereby causing more contact faults, which may not be apparent in other competitions or training situations. A final limitation to the study is that only one single course was observed. The current study does not account for different course designs, which may include different angles of entering the dog walk and different handler positions while the dog is completing the obstacle. These differences could alter paw placement patterns and the speed of obstacle completion. To the knowledge of the authors, the current study is the first to attempt to define paw placement patterns used by canine competitors while completing the dog walk obstacle during agility trials in a large group of dogs while also evaluating differences across height classes.

## Conclusion

In conclusion, the current study determined that there was a high variability in paw placement in dogs performing the dog walk agility obstacle. The study also found large variability in contact behavior of participating canine athletes. Our hypothesis was therefore rejected as there were no easily observed paw placement patterns found. Currently, canine agility is lacking data regarding definition of movement styles and patterns of obstacle completion. It is also unknown how these movement styles and obstacle completion patterns affect long-term consequences for injury risk and the longevity of the canine athlete. Further studies using kinematic and kinetic data collection are required to better define movement patterns and paw placement styles in canine athletes completing the dog walk obstacle. The high variation of paw placement observed in the current study provides further insight into the challenges that come with definition of canine gait styles during activities and the need for further investigation into this field of study. If gait can be more fully assessed, it may be able to be correlated with injury, which may influence training protocols and injury prevention strategies for these athletes.

## Supporting information

**S1 Table.**
(EML)

**S2 Table.**
(EML)

## Acknowledgments

The authors thank additional team members, Amanda Burkhart and Mia Gulan, for their review of a portion of the runs to enable statistical estimate of review consistency.

## Author Contributions

**Conceptualization:** Arielle Pechette Markley, Abigail Shoben.

**Data curation:** Juli K. DiMichele, Abigail Shoben.

**Formal analysis:** Juli K. DiMichele, Arielle Pechette Markley, Nina R. Kieves.

**Investigation:** Arielle Pechette Markley, Abigail Shoben, Nina R. Kieves.

**Methodology:** Juli K. DiMichele, Arielle Pechette Markley, Abigail Shoben, Nina R. Kieves.

**Project administration:** Arielle Pechette Markley, Abigail Shoben, Nina R. Kieves.

**Supervision:** Nina R. Kieves.

**Writing – original draft:** Juli K. DiMichele.

**Writing – review & editing:** Juli K. DiMichele, Arielle Pechette Markley, Abigail Shoben, Nina R. Kieves.

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
