## [Decision Letter · Decision Letter 0]

2 Jan 2024

PONE-D-23-32065Evaluation of variability in performance by dogs completing the dog walk obstacle in agility competitionPLOS ONE

Dear Dr. Kieves,

Thank you for submitting your manuscript to PLOS ONE. After careful consideration, we feel that it has merit but does not fully meet PLOS ONE’s publication criteria as it currently stands. Therefore, we invite you to submit a revised version of the manuscript that addresses the points raised during the review process (included below).

We look forward to receiving your revised manuscript.

Kind regards,

Cord M. Brundage, D.V.M., Ph.D.

Academic Editor

PLOS ONE

2. We notice that your supplementary tables are included in the manuscript file. Please remove them and upload them with the file type 'Supporting Information'. Please ensure that each Supporting Information file has a legend listed in the manuscript after the references list.

Reviewers' comments:

Reviewer's Responses to Questions

**Comments to the Author**

1. Is the manuscript technically sound, and do the data support the conclusions?

Reviewer #1: Yes

Reviewer #2: Yes

2. Has the statistical analysis been performed appropriately and rigorously? 

Reviewer #1: No

Reviewer #2: Yes

3. Have the authors made all data underlying the findings in their manuscript fully available?

Reviewer #1: Yes

Reviewer #2: Yes

4. Is the manuscript presented in an intelligible fashion and written in standard English?

Reviewer #1: Yes

Reviewer #2: Yes

5. Review Comments to the Author

Reviewer #1: 1. Abstract: Please provide more information about the method!

2. Abstract: Did you compare the date statistically? Did you find significant differences?

3. Introduction: Please provide a little information about the motion analysis of canine with different situations (treadmill, unleashed, leashed)

4. Materials and methods: Why does not have camera on the left side?

5. Materials and methods: How did you detect the initial contact?

6. Materials and methods: How did you measure the height of the withers?

7. Materials and methods: Please provide more information about the statistics!

8. Table 1: p for differences? How did you determine?

9. Table 1: I recommend to move the columns All fogs before the columns O for differences?

10. Discussion: Please repeat the aim of the present research in the first paragraphs (before the hypothesis)

11. Conclusion: Please provide the main contribution, the novelty of the present study.

Reviewer #2: Thank you for your manuscript on performance and paw placement pattern of agility dogs completing the dog walk obstacle. I think these types of studies are important as an initial assessment to help guide further kinetic and kinematic studies, as you suggest. Please find my comments for areas I would like clarification as well as some typographical errors I spotted:

Title: I think the title could be made clearer by adding the phrase “and paw placement patterns” after the word “performance” and saying “in an agility competition” since this is looking at one specific competition/course and not across multiple courses or runs.

Line 29-30: Word missing - “as well associated agility-related injuries.” Should read “as well as associated agility-related injuries.”

Line 35: Missing the word “as” here again - “ as well risk of acute injuries while performing...”

Line 110: Typo “Similarly, the first paw to contact each ramp each ramp was also recorded...” the phrase ‘each ramp’ is written twice.

Results – can you please note how many total videos were analyzed? I am interested to know if the 296 attempts represents all dogs/videos or what number of dogs did not even attempt the obstacle (I see these numbers in the tables later but would be nice to see them in the text). I’d also be interested to know if those dogs that did not attempt the obstacle refused that specific obstacle and therefore that is why it was not attempted, or if there was a reason unrelated to the dog walk that this particular obstacle was not attempted.

I would like to know how the remaining dogs approached the dog walk (as a reader I can guess that the remaining 4% of dogs took the obstacle at the wrong time and that 5% took it with the handler to the right side of the obstacle, but I’d appreciate the clarification, especially if this is not the correct assumption). Where these included in the statistical analysis? I ask because I wonder if the paw strike could be influenced by whether the dog walk was approached in the correct order (from the same orientation after the u-tunnel) and which side the handler was on, and suggest that perhaps these should not be included in the stats.

I would suggest that the number of dogs in each height category be briefly discussed in the text of the results section as currently it is only available in the tables.

Line 144: You mention that the down ramp initial contact could not always be observed in the video. For how many dogs was the down ramp fully observed? (I see this in Table 2 now, but I think it would be nice to include this in the text).

Line 144-145: I am finding this sentence to be a little confusing. I think where I become a little confused by the wording is “...again there is a difference by...” specifically the word again, since what you next say is contradictory to your prior statement rather than in agreement (and thus the word ‘again’ is leading me to be a little confused). I think clarity could be improved by removing the word ‘again’ and starting the remainder of this sentence as a new sentence.

I would consider an additional limitation that this is only looking at a single course. Different course designs (different obstacles before and after dog walk or different angles of entering the obstacle) could alter paw placement patterns and speed of obstacle completion.

6. PLOS authors have the option to publish the peer review history of their article (what does this mean?). If published, this will include your full peer review and any attached files.

Reviewer #1: No

Reviewer #2: **Yes: **Christina Montalbano

---

## [Author Response · Author response to Decision Letter 0]

26 Jan 2024

Reviewer #1: 

1. Abstract: Please provide more information about the method!

Author Response: We have added some clarification to the abstract to indicate that this was an online video review to clarify the methodology. The primary statistical analysis was descriptive, so we have not specifically included that in the abstract. However, please let us know if there are additional details of our methods that should be included in the abstract. (Lines 16-19) 

2. Abstract: Did you compare the date statistically? Did you find significant differences?

Author Response: We assume this comment refers to the comparisons between height classes referenced in the abstract. As noted in the text of our methods, our primary objective of this study was descriptive, so our results in the abstract reflect this descriptive goal. We have added the word “qualitatively” to indicate this in the abstract. (Line 19)

3. Introduction: Please provide a little information about the motion analysis of canine with different situations (treadmill, unleashed, leashed)

Author Response: The authors are unsure what additional information the reviewer feels would be important to include in the introduction regarding motion analysis. If you could clarify the authors would be happy to consider amending the introduction in future versions of the manuscript. 

4. Materials and methods: Why does not have camera on the left side?

Author Response: Thank you for this question. This study was done using a recording of the event created by 4LeggedFlix. We did not select the camera placement and utilized the one view that was available for this study. We have added clarification of this point to the text. (Lines 94-95)

5. Materials and methods: How did you detect the initial contact?

Author Response: Initial contact with the obstacle was observed by video review while watching the video at reduced speed. We have added clarification to the text regarding this. (Line 100)

6. Materials and methods: How did you measure the height of the withers?

Author Response: The height of the withers was inferred from the dog’s jump height class measured and provided by the UKI program directory that was available online. We have clarified this point in the text. (Lines 128-130)

7. Materials and methods: Please provide more information about the statistics!

Author Response: We have added additional clarification of how p for difference was determined for the time variables in table 1 (lines 137-139). We otherwise believe the description of the statistics used is complete. 

8. Table 1: p for differences? How did you determine?

Author Response: These were multivariate Wald tests following linear regression with robust standard errors. This has been clarified in the methods. (Lines 137-139)

9. Table 1: I recommend to move the columns All fogs before the columns O for differences?

Author Response: Thank you for this suggestion. Based on this suggestion, we have moved the “all dogs” to the first column for readability. 

10. Discussion: Please repeat the aim of the present research in the first paragraphs (before the hypothesis)

Author Response: Thank you for the suggestion. A sentence was added to the discussion to clarify the aim of the study. (Lines 234-235)

11. Conclusion: Please provide the main contribution, the novelty of the present study.

Author Response: Thank you for the suggestion. We have added text to the conclusion to provide the main contribution and importance of the study. (Lines 350-353)

Reviewer #2: Thank you for your manuscript on performance and paw placement pattern of agility dogs completing the dog walk obstacle. I think these types of studies are important as an initial assessment to help guide further kinetic and kinematic studies, as you suggest. Please find my comments for areas I would like clarification as well as some typographical errors I spotted:

Title: I think the title could be made clearer by adding the phrase “and paw placement patterns” after the word “performance” and saying “in an agility competition” since this is looking at one specific competition/course and not across multiple courses or runs.

Author Response: Thank you for the suggestion, this change has been made.

Line 29-30: Word missing - “as well associated agility-related injuries.” Should read “as well as associated agility-related injuries.”

Author Response: Thank you for the careful read and noticing this error. This has been corrected. (Line 30) 

Line 35: Missing the word “as” here again - “ as well risk of acute injuries while performing...”

Author Response: Thank you for noticing this error, it has been corrected. (Line 36) 

Line 110: Typo “Similarly, the first paw to contact each ramp each ramp was also recorded...” the phrase ‘each ramp’ is written twice. 

Author Response: Thank you for noticing this error, it has been corrected. (Line 112) 

Results – can you please note how many total videos were analyzed? I am interested to know if the 296 attempts represents all dogs/videos or what number of dogs did not even attempt the obstacle (I see these numbers in the tables later but would be nice to see them in the text). I’d also be interested to know if those dogs that did not attempt the obstacle refused that specific obstacle and therefore that is why it was not attempted, or if there was a reason unrelated to the dog walk that this particular obstacle was not attempted.

Author Response: Thank you for these thoughtful comments. All videoed runs of the course were reviewed; however, we did not specifically capture if a dog started the course, but did not attempt the dog walk obstacle. We believe this was rare, but do not have the exact data to provide. The 296 represents all dogs who attempted the course and also attempted the dog walk obstacle (defined as putting at least one paw on the obstacle). We have reworded the first line of the results to help clarify. (Lines 153-154)

I would like to know how the remaining dogs approached the dog walk (as a reader I can guess that the remaining 4% of dogs took the obstacle at the wrong time and that 5% took it with the handler to the right side of the obstacle, but I’d appreciate the clarification, especially if this is not the correct assumption). Where these included in the statistical analysis? I ask because I wonder if the paw strike could be influenced by whether the dog walk was approached in the correct order (from the same orientation after the u-tunnel) and which side the handler was on, and suggest that perhaps these should not be included in the stats.

Author Response: Thank you for the suggestion. We have added the information about approaching the dog walk in flow and explicitly about handler side to the results (lines 155-157). We agree that paw strike could be influenced by approach (although here all but two dogs took the u-shaped tunnel first) and by handler side. However, our goal was to describe the patterns observed completing the obstacle within a course; not specifically to describe patterns observed with a prescribed approach and prescribed handler side so our primary analysis included all dogs. We note that restricting the analysis to dogs who took the dog walk in flow and with the handler on the left (n=270) showed very similar patterns to the full sample. We have added to the discussion the idea that handler position and approach may impact paw strike patterns and completion times. (Lines 335-338)

I would suggest that the number of dogs in each height category be briefly discussed in the text of the results section as currently it is only available in the tables.

Author Response: We have added text to the results to include this information. (Lines 161-163) 

Line 144: You mention that the down ramp initial contact could not always be observed in the video. For how many dogs was the down ramp fully observed? (I see this in Table 2 now, but I think it would be nice to include this in the text).

Author Response: Thank you for the suggestion. We have added the number of dogs for whom the down ramp initial contact was observed (262) to the text. (Line 169)

Line 144-145: I am finding this sentence to be a little confusing. I think where I become a little confused by the wording is “...again there is a difference by...” specifically the word again, since what you next say is contradictory to your prior statement rather than in agreement (and thus the word ‘again’ is leading me to be a little confused). I think clarity could be improved by removing the word ‘again’ and starting the remainder of this sentence as a new sentence.

Author Response: Thank you for the suggestion. A new sentence was created to make this clearer. (Lines 169-170)

I would consider an additional limitation that this is only looking at a single course. Different course designs (different obstacles before and after dog walk or different angles of entering the obstacle) could alter paw placement patterns and speed of obstacle completion.

Author Response: Thank you for the suggestion. A statement was added to the discussion regarding this limitation. (Lines 334-336)

---

## [Decision Letter · Decision Letter 1]

13 Feb 2024

Evaluation of variability in performance and paw placement patterns by dogs completing the dog walk obstacle in an agility competition

PONE-D-23-32065R1

Dear Dr. Kieves,

We’re pleased to inform you that your manuscript has been judged scientifically suitable for publication and will be formally accepted for publication once it meets all outstanding technical requirements.

Kind regards,

Cord M. Brundage, D.V.M., Ph.D.

Academic Editor

PLOS ONE

Reviewers' comments:

Reviewer's Responses to Questions

**Comments to the Author**

1. If the authors have adequately addressed your comments raised in a previous round of review and you feel that this manuscript is now acceptable for publication, you may indicate that here to bypass the “Comments to the Author” section, enter your conflict of interest statement in the “Confidential to Editor” section, and submit your "Accept" recommendation.

Reviewer #1: (No Response)

Reviewer #2: All comments have been addressed

2. Is the manuscript technically sound, and do the data support the conclusions?

Reviewer #1: Yes

Reviewer #2: Yes

3. Has the statistical analysis been performed appropriately and rigorously? 

Reviewer #1: Yes

Reviewer #2: Yes

4. Have the authors made all data underlying the findings in their manuscript fully available?

Reviewer #1: Yes

Reviewer #2: (No Response)

5. Is the manuscript presented in an intelligible fashion and written in standard English?

Reviewer #1: Yes

Reviewer #2: Yes

6. Review Comments to the Author

Reviewer #1: Thank you for your answer, the correction, extension is enough for me. I recommend the corrected version for acceptance.

Reviewer #2: (No Response)

7. PLOS authors have the option to publish the peer review history of their article (what does this mean?). If published, this will include your full peer review and any attached files.

Reviewer #1: No

Reviewer #2: **Yes: **Christina Montalbano

---

## [Editor Report · Acceptance letter]

26 Feb 2024

PONE-D-23-32065R1 

PLOS ONE

Dear Dr. Kieves, 

I'm pleased to inform you that your manuscript has been deemed suitable for publication in PLOS ONE. Congratulations! Your manuscript is now being handed over to our production team.

Kind regards, 

on behalf of

Dr. Cord M. Brundage 

Academic Editor

PLOS ONE